# Assessing the Impact of Long-Term High-Dose Statin Treatment on Pericoronary Inflammation and Plaque Distribution—A Comprehensive Coronary CTA Follow-Up Study

**DOI:** 10.3390/ijms25031700

**Published:** 2024-01-30

**Authors:** Botond Barna Mátyás, Imre Benedek, Nóra Raț, Emanuel Blîndu, Zsolt Parajkó, Theofana Mihăilă, Theodora Benedek

**Affiliations:** 1Clinic of Cardiology, Mureș County Emergency Clinical Hospital, 540136 Târgu Mureș, Romania; matyas_botond@yahoo.com (B.B.M.); imre.benedek@umfst.ro (I.B.); emi.blindu@yahoo.com (E.B.); p_zsolt92@yahoo.com (Z.P.); theofana_m@yahoo.com (T.M.); theodora.benedek@umfst.ro (T.B.); 2Doctoral School of Medicine and Pharmacy, “George Emil Palade” University of Medicine, Pharmacy, Science and Technology of Târgu Mureș, 540139 Târgu Mureș, Romania; 3Department of Cardiology, “George Emil Palade” University of Medicine, Pharmacy, Science and Technology of Târgu Mureș, 540139 Târgu Mureș, Romania

**Keywords:** pericoronary adipose tissue inflammation, computed tomography angiography, fat attenuation index score, atherosclerosis, coronary artery disease, high-dose statin

## Abstract

Computed tomography angiography (CTA) has validated the use of pericoronary adipose tissue (PCAT) attenuation as a credible indicator of coronary inflammation, playing a crucial role in coronary artery disease (CAD). This study aimed to evaluate the long-term effects of high-dose statins on PCAT attenuation at coronary lesion sites and changes in plaque distribution. Our prospective observational study included 52 patients (mean age 60.43) with chest pain, a low-to-intermediate likelihood of CAD, who had documented atheromatous plaque through CTA, performed approximately 1 year and 3 years after inclusion. We utilized the advanced features of the CaRi-Heart^®^ and syngo.via Frontier^®^ systems to assess coronary plaques and changes in PCAT attenuation. The investigation of changes in plaque morphology revealed significant alterations. Notably, in mixed plaques, calcified portions increased (*p* < 0.0001), while non-calcified plaque volume (NCPV) decreased (*p* = 0.0209). PCAT attenuation generally decreased after one year and remained low, indicating reduced inflammation in the following arteries: left anterior descending artery (LAD) (*p* = 0.0142), left circumflex artery (LCX) (*p* = 0.0513), and right coronary artery (RCA) (*p* = 0.1249). The CaRi-Heart^®^ risk also decreased significantly (*p* = 0.0041). Linear regression analysis demonstrated a correlation between increased PCAT attenuation and higher volumes of NCPV (*p* < 0.0001, r = 0.3032) and lipid-rich plaque volume (*p* < 0.0001, r = 0.3281). Our study provides evidence that high-dose statin therapy significantly reduces CAD risk factors, inflammation, and plaque vulnerability, as evidenced by the notable decrease in PCAT attenuation, a critical indicator of plaque progression.

## 1. Introduction

Coronary artery disease (CAD) is a major global health challenge and the leading cause of death worldwide, responsible for over nine million deaths in 2019 [1,2]. It manifests in various forms, such as stable and unstable angina, myocardial infarction (MI), and sudden cardiac death [3]. Significantly, a quarter of MI patients develop heart failure, which has a high five-year mortality rate of 50%, placing considerable strain on healthcare systems [4].

Pericoronary adipose tissue (PCAT) is a specific type of epicardial adipose tissue (EAT) that surrounds the coronary arteries, with distinct morphological and functional characteristics despite its close proximity to EAT [5,6]. Emerging research suggests that PCAT plays a unique role in cardiovascular diseases, making both its quantitative and qualitative assessment crucial for evaluating an individual’s risk of cardiometabolic disorders [5,7,8]. Vascular inflammation within PCAT is a hallmark of unstable “vulnerable” coronary plaques and a key contributor to the progression of coronary atherosclerosis [9,10,11]. Utilizing computed tomography angiography (CTA) to measure the fat attenuation index (FAI) has become clinically important for identifying high-risk patients [12,13] and assessing the impact of anti-inflammatory treatments, such as how statins have been shown to reduce the CT attenuation in EAT, indicating their therapeutic effect [14].

CTA serves as a noninvasive method for evaluating PCAT, using the FAI to monitor changes in PCAT and track coronary plaque progression. An increase in PCAT density often signifies various stages of CAD [13] and is associated with vulnerable plaque characteristics, predictors of clinical incidents [15,16], especially in acute coronary syndrome [17]. However, the study of PCAT’s role in plaque progression is still emerging, limited by patient numbers and follow-up duration, leaving the full impact of PCAT density changes on plaque progression yet to be fully understood [18,19].

Statin therapy plays a crucial role in stabilizing plaques in CAD, mainly by reducing low-density and fibro-fatty plaque volumes and increasing denser, calcium-rich plaques. This increase in calcium in heart arteries further lowers plaque detachment risk [20]. Statins work through various mechanisms, including reducing lipid accumulation in plaques, decreasing inflammation, and enhancing endothelial function [21,22]. They effectively decrease plaque and external elastic membrane volumes without impacting the lumen volume, primarily due to their anti-inflammatory properties. However, there are some associated risks with intensive statin use [23].

Against this backdrop, our objective was to investigate the effects of extended high-dose statin therapy on the FAI of PCAT at coronary lesion sites. Additionally, we sought to assess alterations in plaque distribution over time, utilizing a 128-slice contrast-enhanced CTA during follow-up appointments.

## 2. Results

### 2.1. Baseline Characteristics of the Study Population

In this study, we enrolled 52 participants who met all the previously outlined inclusion criteria. The average age of these patients at the initial scan was 60.43 ± 9.21 years, with males constituting 65.38% (*n* = 34) of the cohort. The cohort had a high prevalence of traditional risk factors for CAD, with hypertension being the most prevalent at 84.61% (*n* = 44). Additionally, 63.46% (*n* = 33) of the patients had hyperlipidemia, while diabetes mellitus and smoking were present in 26.92% (*n* = 14) and 17.30% (*n* = 9) of the patients, respectively. A considerable number of patients, accounting for 42.30% (*n* = 22), indicated a family history of CAD, underscoring the hereditary aspect of the disease in their familial background. The composition of plaque types within the selected coronary arteries was categorized as follows: 54.06% (*n* = 80) were calcified plaques, 4.73% (*n* = 7) were non-calcified plaques, and 41.21% (*n* = 61) were mixed plaques featuring both calcified and non-calcified components. The mean calcium score among the participants was 127.5 ± 72.96. The distribution across different calcium score ranges was as follows: scores < 10 were observed in 5.77% (*n* = 3), scores ranging from 10 to 400 comprised the majority at 69.23% (*n* = 36), and scores > 400 were found in 25.00% (*n* = 13). These findings are presented in Table 1.

### 2.2. Serial Changes in the Lipid Panel Outcomes during Follow-Up

After commencing statin therapy, the blood tests conducted during the first-year follow-up revealed significant changes in lipid profiles. The average total cholesterol levels decreased significantly from 194.3 ± 66.76 to 145.2 ± 34.7 (*p* = 0.0003). Notable alterations were also observed in LDL-Cho and HDL-Cho levels, with LDL-Cho decreasing from 105.9 ± 33.97 to 87.69 ± 32.99 (*p* < 0.0001) and HDL-Cho increasing from 38.11 ± 8.72 to 47.10 ± 7.97 (*p* < 0.0001).

Long-term analysis at the time of the last visit revealed persistent significant differences in these parameters when comparing initial and final visits, with total cholesterol (*p* < 0.0001), LDL-Cho (*p* < 0.0001), and HDL-Cho (*p* < 0.0001) all showing notable differences. Triglyceride levels also changed significantly after one year and showed a significant long-term decrease from 188.7 ± 66.01 to 171.9 ± 53.76 (*p* < 0.0001), reflecting effective anti-triglyceride therapy. These findings are detailed in Table 2.

### 2.3. Serial Changes of Plaque Features before and after Statin Treatment

Subsequently, we examined the morphological changes in plaque features during the follow-up period. However, due to the minimal changes experienced during the short (1 year) follow-up and the very time-consuming evaluation procedure, we only performed plaque analysis on the baseline CTA scans and the last multi-year control scans. The study observed an overall increase in the volume of calcified plaques, though this change was not statistically significant between the initial and final follow-up CTA scans (*p* = 0.0773). Regarding non-calcified plaques, changes were also noted, highlighting the dynamic and variable nature of plaque morphology. However, only seven plaques were analyzed in this category, and a significant transformation was observed solely in the CPV over a period of 3 years. 

Figure 1 presents the TPV values for each plaque type, illustrating the shifts in plaque morphology during the follow-up period. However, these changes were not statistically significant.

In our longitudinal analysis of plaque morphology, we found the mixed plaques particularly intriguing due to their dynamic changes. Specifically, the CPV within these plaques significantly increased from 53.92 ± 31.29 mm^3^ to 87.42 ± 43.48 mm^3^ (*p* < 0.0001). In contrast, the NCPV significantly decreased from 180.5 ± 66.81 mm^3^ to 155.4 ± 59.51 mm^3^ (*p* = 0.0209). Although the TPV increased, this change was not statistically significant, moving from 237.4 ± 70.0 mm^3^ to 256.9 ± 79.84 mm^3^ (*p* = 0.1454). Further analysis of the NCPV showed an increase in the volume of fibrotic plaques from 154.8 ± 63.04 mm^3^ to 179.8 ± 63.46 mm^3^ (*p* = 0.0324), while lipid-rich plaques significantly decreased from 20.94 ± 9.93 mm^3^ to 16.62 ± 7.69 mm^3^ (*p* = 0.0057). In summary, during the follow-up period, there was a decrease in NCPV with significant changes in the FPV and LRPV, alongside a significant increase in CPV in mixed plaques. These findings are further detailed in Table 3 and Figure 2.

The findings we have discussed are best validated by a specific instance from our patient pool, which is depicted in detail in Figure 3. This illustration provides a clear view of how statin therapy contributes to the reduction in FAI score and the alteration of plaque characteristics, showcasing the changes in a patient over the course of treatment.

### 2.4. Serial Changes of Lesion-Specific PCAT-FAI before and after Statin Treatment

For all lesions across the coronary arteries, follow-up data revealed that high-dose statin therapy significantly altered the lesion-specific PCAT-FAI after nearly one year, with a general trend of gradual decrease remaining thereafter at the last scan. Figure 4 displays the diagrams that break down the data into the two specific follow-up intervals: 1 year and over 3 years. The specific details for all the PCAT-FAI parameters are outlined separately.

It is important to note that the traditionally measured FAI in HU units showed a significant decrease only in the LAD at the one-year follow-up (−68.94 ± 6.88 vs. −72.83 ± 6.29, *p* = 0.0061). This reduction remained consistent up to the final scans (−68.94 ± 6.88 vs. −71.75 ± 8.07, *p* = 0.0138).

Regarding specific FAI scores, there was an overall decrease across all three coronary arteries at the one-year mark compared to initial measurements. This decrease was significant for the total score (16.78 ± 8.76 vs. 12.05 ± 7.88, *p* < 0.0001), LAD (15.93 ± 9.22 vs. 11.75 ± 7.35, *p* = 0.0109), LCX (14.78 ± 7.35 vs. 10.80 ± 7.27, *p* = 0.0029), and RCA (19.60 ± 8.99 vs. 13.13 ± 8.34, *p* < 0.0001). Although these values started to increase slightly by the time of the last CTA, a significant difference was maintained for the total score (16.78 ± 8.76 vs. 13.64 ± 8.00, *p* = 0.0007), LAD (15.93 ± 9.22 vs. 12.03 ± 6.27, *p* = 0.0142), and LCX (14.78 ± 7.35 vs. 12.15 ± 7.51, *p* = 0.0513). However, for the RCA, after 3 years, the FAI score increased to the point where the difference was no longer significant (19.60 ± 8.99 vs. 16.73 ± 9.17, *p* = 0.1249).

The FAI scores for the LAD, LCX, and RCA were plotted on percentile curves for different age and sex groups, with their predictive value assessed using Cox proportional hazards models. These models were adjusted for risk factors such as hypertension, diabetes, smoking, hyperlipidemia, high-risk plaque features, and the modified Duke CAD prognostic index [24].

Concurrently, a notable shift in the PCAT-FAI score percentiles was observed, indicating a significant decrease across all three coronary arteries when comparing baseline scans with subsequent scans. For the LAD, scores decreased from 72.88 ± 16.22 to 64.96 ± 24.45 (*p* = 0.0526) and further to 61.91 ± 17.96 (*p* = 0.0044). The LCX scores dropped from 73.56 ± 15.60 to 63.85 ± 16.32 (*p* = 0.0120) and then to 60.18 ± 18.07 (*p* < 0.0001). For the RCA, the scores went from 81.73 ± 13.61 to 70.73 ± 18.98 (*p* = 0.0001) and subsequently changed to 71.72 ± 19.21 (*p* = 0.0041). 

There was a notable reduction in the CaRi-Heart^®^ Risk score between the initial and second scans (33.20 ± 22.07 vs. 17.78 ± 13.41, *p* = 0.0001). Although there was an increase in the score at the time of the last visit, it still remained significantly lower compared to the initial value (33.20 ± 22.07 vs. 20.65 ± 16.14, *p* = 0.0041), as depicted in Figure 5. The CaRi-Heart^®^ analysis takes into account age and gender to assess coronary inflammation and predicts the 8-year risk of a fatal cardiac event. This evaluation includes the FAI Score, plaque burden, and various clinical risk factors.

### 2.5. Relationship between the PCAT-FAI Assessment and Plaque Component Morphology

Considering the individual PCAT-FAI score values for each plaque, we conducted a linear regression analysis based on the baseline CTA scans. Intriguingly, this shed light on what the literature has previously speculated but not studied with FAI scoring: an increase in FAI score correlates with an increase in TPV (*p* = 0.0569, r = 0.0600), NCPV (*p* < 0.0001, r = 0.3032), and significantly, with an increase in LRPV (*p* < 0.0001, r = 0.3281). However, such a correlation was not found between the FAI score and the TPV of calcified plaques (*p* = 0.4277), nor in the CPV (*p* = 0.2021) or FPV (*p* = 0.9992) of mixed plaques (Figure 6). 

## 3. Discussion

Our study’s main outcome strongly supports the benefits of statin therapy in reducing risk factors and inflammation associated with CAD. The data from our participant cohort, demonstrating significant reductions in lipid levels, plaque-specific inflammation, and changes in plaque structure, indicate that high-dose statin therapy has a positive impact on both the biochemical and structural aspects of CAD pathology.

The FAI score correlates with the vulnerable component of plaque, a topic that recent studies have investigated, particularly examining the relationship between PCAT-FAI and the increase in NCPV and LRPV. These studies have focused on the predictive power of PCAT attenuation and NCPV concerning MI. Research has demonstrated that PCAT attenuation around the RCA can predict MI occurrence, making it a significant predictive marker for assessing the 5-year MI risk [25]. Additionally, the combination of FAI and plaque assessments has been shown to more effectively distinguish ischemia compared to evaluations based solely on stenosis. This highlights the significance of FAI as a marker for the risk of coronary atherosclerosis, supporting its strong association with the development of coronary atherosclerosis and plaque vulnerability [26,27].

Statins not only improved lipid profiles but also seemed to induce a shift in plaque morphology, with increases in CPV suggesting a move towards more stable plaque forms, and a decrease in NCPV reflecting potential reductions in vulnerability to rupture.

This study’s findings resonate with existing research that links statin use with a slowdown in CAD progression and improved lipid regulation [28]. It is well-established that statins can notably decrease the advancement of atherosclerotic plaques and the incidence of MACE, despite increasing coronary calcium, which may be indicative of plaque stabilization [29,30,31]. The relevance of high-dose statins in delivering additional vascular protection above standard doses is supported by clinical trials, though the appropriateness of this therapy remains patient-specific due to the variability in clinical conditions and risks [32,33]. Such therapy’s impact on the different components of coronary plaques has also been substantiated, supporting the study’s report of differential changes in plaque types [19,34].

Integrating recent advancements in CTA into our study enriches the understanding of statin therapy in combating CAD. The use of radiomics and radio-transcriptomics for detecting vulnerable plaques and forecasting potential cardiac events in both heart disease and COVID-19 patients complements our findings about the role of statins in stabilizing plaques. This highlights statins’ dual role in both managing lipid levels and modifying plaque composition to reduce instability. Adopting these sophisticated imaging methods could deepen insights into the effectiveness of statin treatment, paving the way for more tailored therapeutic approaches in CAD management [35,36].

Nonetheless, this research is constrained by factors such as its limited participant number and relatively brief follow-up period, which may not fully reflect the long-term effects of statin treatment. The potential for selection bias given by the sole use of CTA scans for plaque evaluation also poses a limitation, as some plaques may not be detectable by this method alone. The participant selection, characterized by a high prevalence of traditional CAD risk factors, could additionally limit the generalizability of the study’s results. Future research would benefit from a broader and more diverse cohort and an extended monitoring timeline to validate the findings presented here. Utilizing a variety of imaging techniques could offer a more nuanced view of the changes in plaque composition over time.

In summary, the clinical implications of this study are significant, underlining the importance of high-dose statin therapy in the management and potential modification of CAD. The evidence of statin efficacy in improving lipid profiles, reducing plaque-specific inflammation, and stabilizing plaque morphology holds considerable promise for CAD treatment strategies. Although the study’s limitations indicate a need for ongoing research, they do not undermine the current clinical application of its findings. Future studies, addressing these limitations, could provide even stronger support for the use of statins in CAD management and prevention strategies.

## 4. Materials and Methods

### 4.1. Study Design and Population

In our single-center, longitudinal, prospective, observational, nonrandomized study, 109 patients with chest pain and a low to intermediate risk of CAD were recruited. These participants underwent a detailed examination of their coronary anatomy, atherosclerotic changes, FAI scoring, and plaque analysis using 128-slice CTA. The study was designed to monitor the same group of individuals over time to observe any developments in their condition. Follow-up periods were divided into two intervals: one spanning approximately one year, and another extending beyond three years.

Our study applied the following inclusion criteria: (1) the initial CTA had to show at least one lesion with a luminal stenosis of 25–49% in any major coronary artery with a diameter of ≥2 mm, below CAD-RADS category 3; (2) participants had to be new to statin therapy and free from any other lipid-lowering medications at the time of their baseline scan. The following exclusion criteria were applied: (1) patients whose coronary CTA results were not clear enough for quantitative assessment (*n* = 23) or PCAT density analysis (*n* = 15) at either the initial or follow-up scans, precluding good quality analysis; (2) patients who had clinical incidents leading to coronary revascularization in the interval between their CTAs (*n* = 19). After applying these exclusion criteria, the final analysis included 148 lesions from 52 patients (as shown in Figure 7).

At each visit, comprehensive lipid profiles, including measurements of total cholesterol, triglycerides, low-density lipoprotein cholesterol (LDL-Cho), and high-density lipoprotein cholesterol (HDL-Cho) levels, were collected. Additionally, we meticulously recorded demographic data, other laboratory test results, cardiovascular risk factors, and any new or progressing symptoms for each participant.

### 4.2. Coronary CTA Acquisition Protocol

All study procedures utilized a 128-slice scanner (Somatom Definition AS, Siemens Healthineers, Erlangen, Germany) for conducting CTA scans on all study participants. Patients with heart rates under 65 bpm underwent scans with retrospective gating and specific technical settings. If heart rates exceeded 65 bpm, beta-blockers were used to achieve the target heart rate, with continuous blood pressure monitoring. The procedure included a non-contrast coronary calcium scan, followed by an iodine contrast injection and a saline flush, with patients holding their breath. The scans were systematically archived in a specialized electronic imaging database, which facilitated offline image post-processing and cloud-based distribution.

### 4.3. Pericoronary Adipose Tissue FAI and Plaque Analysis

Images obtained from the scans were converted to DICOM format, anonymized, and then securely sent to our collaborative research facility, the Centre of Caristo Diagnostics in Oxford, UK, for post-processing. This process involved analyzing inflammation in the PCAT for each coronary artery. Utilizing advanced artificial intelligence (AI) algorithms from CaRi-Heart^®^ by Caristo Diagnostics (Centre of Caristo Diagnostics, Oxford, UK), the PCAT-FAI and AI-based FAI scores were accurately calculated for each major coronary artery across all patients.

The AI algorithms enhance the precision of the PCAT-FAI by measuring attenuation in 3D layers, each 1 mm thick, around the coronary arteries [6,37]. These algorithms perform a series of complex operations, including segmenting cardiac structures and evaluating the PCAT. They adjust for a range of scan-related variables to ensure that the PCAT-FAI is a reliable marker of coronary inflammation, distinct from standard CT attenuation values [38]. 

To clarify, the distinctions between FAI, FAI score, and CaRi-Heart^®^ risk are as follows: (1) FAI, measured in Hounsfield units (HU), offers an unmodified graphical depiction of inflammation levels in the three primary epicardial coronary arteries; (2) FAI score provides a customized evaluation, quantifying coronary inflammation in these arteries while incorporating age and gender, and is expressed as a relative risk; (3) CaRi-Heart^®^ risk represents the absolute risk of experiencing a fatal cardiac event within the next eight years. This risk assessment is based on individualized FAI score values, the extent of coronary atherosclerotic plaque, and other clinical risk factors (diabetes, smoking, hyperlipidemia, and hypertension) [39]. 

For the assessment of plaque characteristics and components at baseline and during follow-up visits, we utilized the Syngo.via Frontier^®^ (Syngo.Via, Siemens Healthineers, Erlangen, Germany) offline workstation. This enabled us to measure various types of plaque volumes, including total (TPV), calcified (CPV), non-calcified (NCPV), lipid-rich (LRPV), and fibrotic (FPV) plaques. Furthermore, we visually categorized all target lesions into one of three predefined types: calcified, non-calcified, or mixed (containing both calcified and non-calcified components).

### 4.4. Statistical Analysis

Following the calculation of the PCAT-FAI for each coronary artery, the results were transmitted to our institution and cataloged in a Microsoft Excel electronic database. For statistical evaluation, we used GraphPad Prism 9.5 software (GraphPad Software, Inc., San Diego, CA, USA). Our analysis encompassed PCAT-FAI measurements across 148 coronary arteries, including 50 from the LAD, 48 from the LCX, and 50 from the RCA. We also calculated the CaRi-Heart^®^ risk scores for each participant.

Data from various follow-up periods were comparatively analyzed. We displayed categorical data as counts and percentages, applying the Chi-square test or Fisher’s exact test for analysis, contingent on dataset dimensions. For continuous data, we provided means ± standard deviations, employing the paired Student’s *t*-test for normally distributed datasets, or the Wilcoxon signed-rank test for those with skewed distributions. To explore the association between shifts in PCAT-FAI values and alterations in plaque attributes, Pearson’s correlation was used for normally distributed variables, while Spearman’s rank correlation was utilized for others. A *p* < 0.05 was used to denote statistical significance.

## 5. Conclusions

Our study validates the efficacy of high-dose statin therapy in diminishing risk factors, inflammation, and plaque vulnerability in CAD. It demonstrates that a high FAI score, indicative of inflamed PCAT and vulnerable plaque morphology, is significantly reduced following high-dose statin treatment. This underscores the significance of the FAI score as a predictive marker for the progression and evolution of CAD.

## Figures and Tables

**Figure 1 ijms-25-01700-f001:**
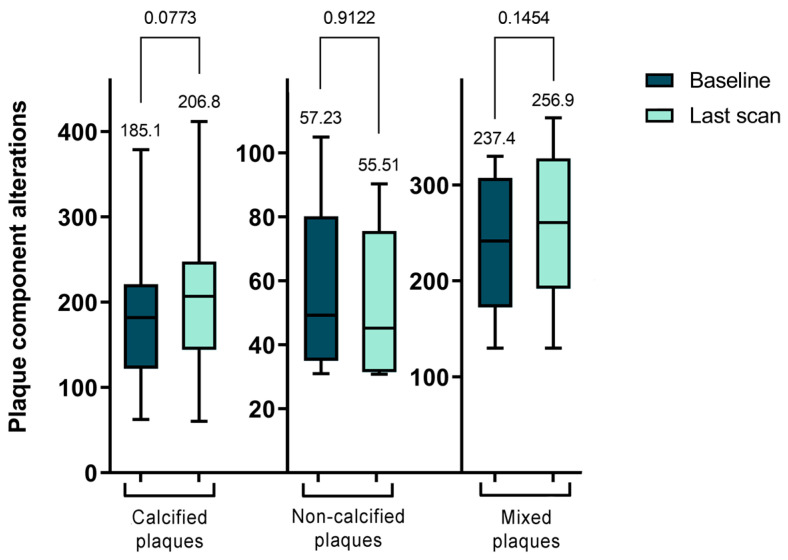
Serial changes of total plaque volume (TPV) for each plaque type.

**Figure 2 ijms-25-01700-f002:**
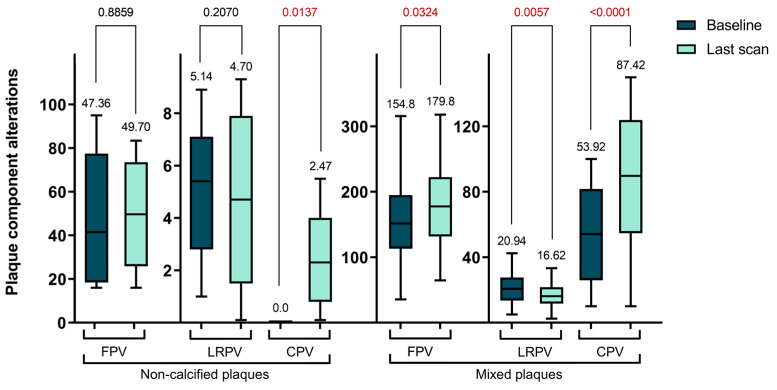
Serial changes of plaque components in the non-calcified and mixed plaque types (FPV—fibrotic plaque volume; LRPV—lipid-rich plaque volume; CPV—calcified plaque volume).

**Figure 3 ijms-25-01700-f003:**
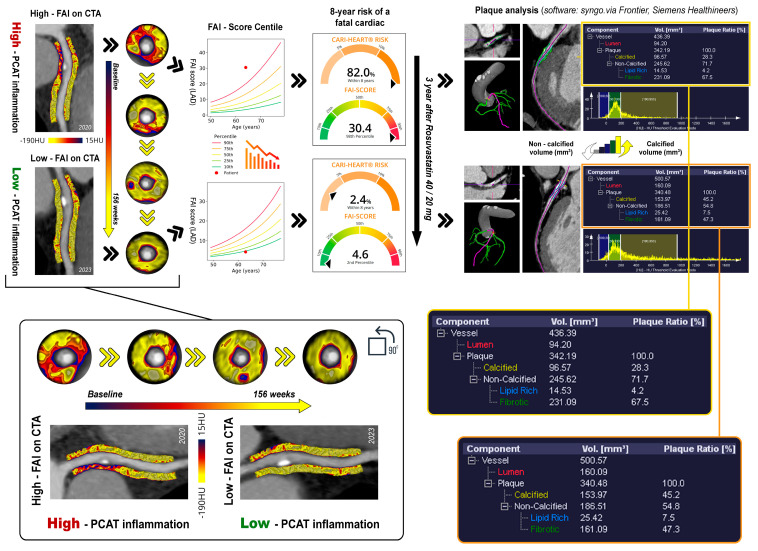
A schematic illustration of a patient case showing a high baseline FAI score with subsequent reduction in vascular inflammation after three years of statin therapy, along with an analysis of changes in plaque composition over the same follow-up period (PCAT—pericoronary adipose tissue; FAI—fat attenuation index; CTA—computed tomography angiography; HU—Hounsfield units; LAD—left anterior descending artery).

**Figure 4 ijms-25-01700-f004:**
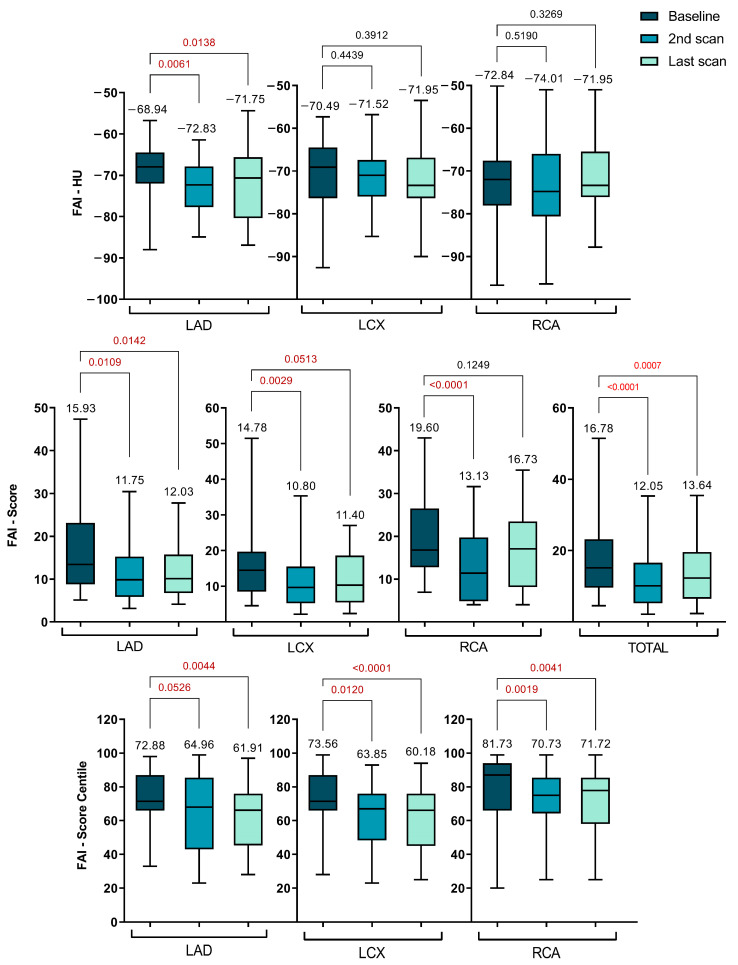
Serial changes of PCAT-FAI before and after statin treatment during the scans (FAI—fat attenuation index; HU—Hounsfield units; LAD—left anterior descending artery; LCX—left circumflex artery; RCA—right coronary artery).

**Figure 5 ijms-25-01700-f005:**
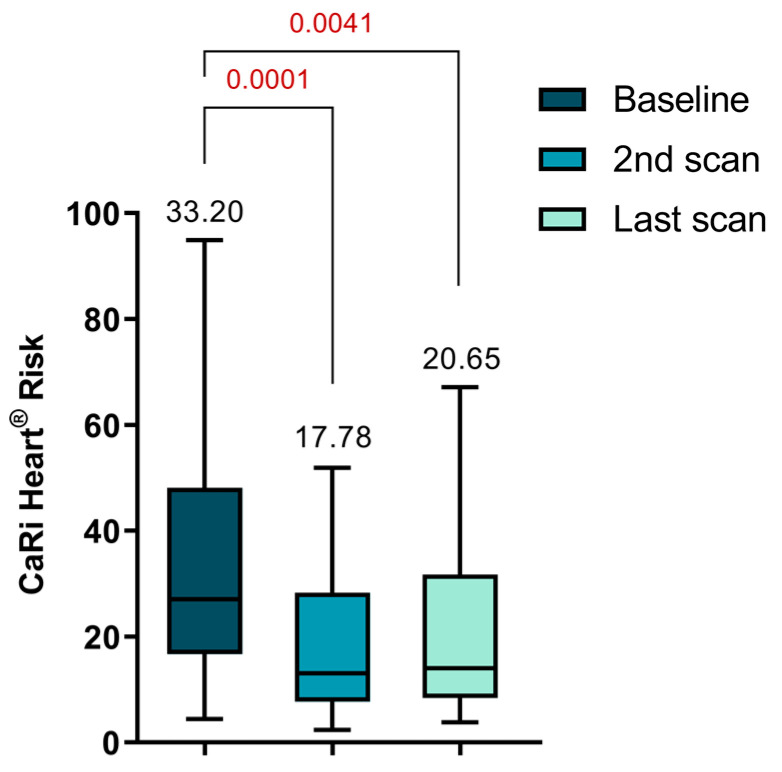
Personalized CaRi-Heart^®^ cardiovascular risk assessment.

**Figure 6 ijms-25-01700-f006:**
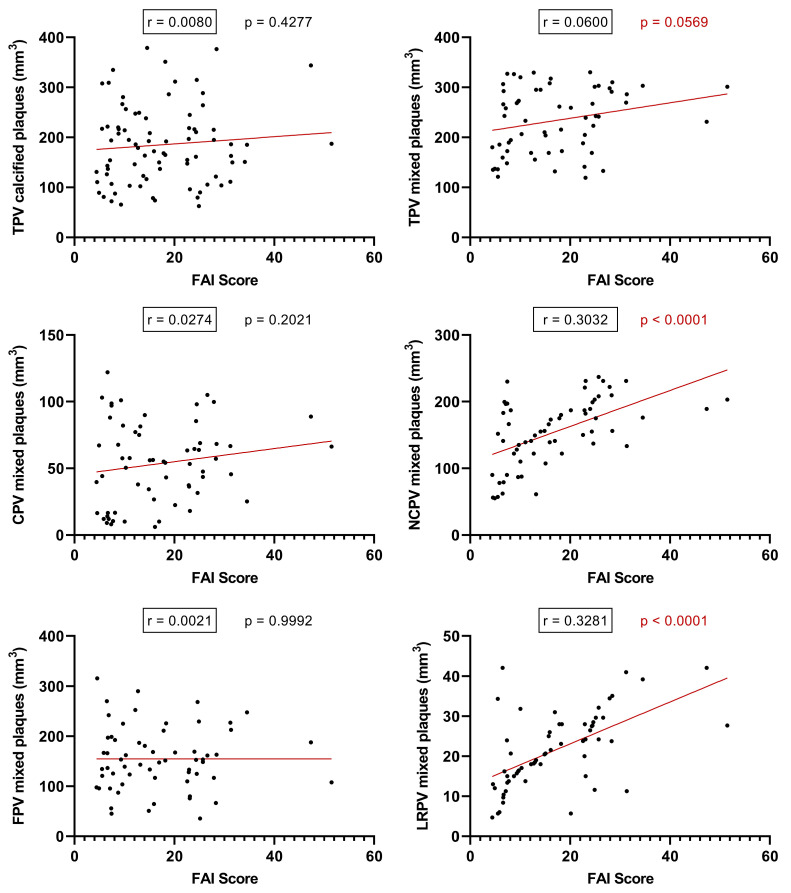
Analyzing PCAT-FAI score in relation to plaque components using linear regression (TPV—total plaque volume; CPV—calcified plaque volume; FPV—fibrotic plaque volume; NCPV—non-calcified plaque volume; LRPV—lipid-rich plaque volume; FAI—fat attenuation index).

**Figure 7 ijms-25-01700-f007:**
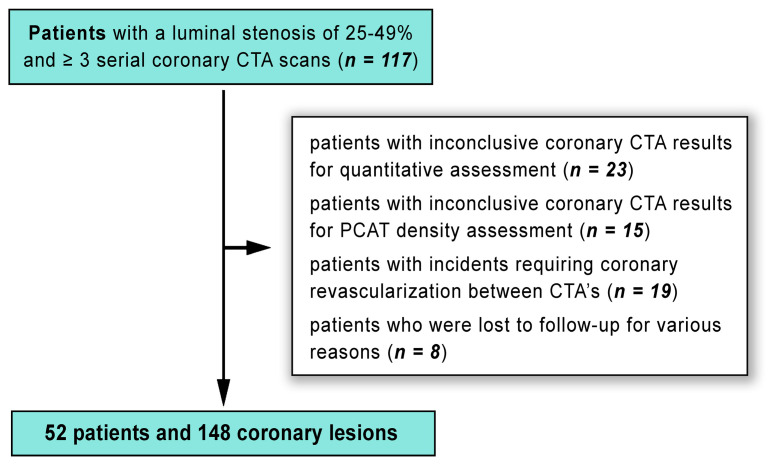
Patient recruitment flowchart (CTA—computed tomography angiography; PCAT—pericoronary adipose tissue).

**Table 1 ijms-25-01700-t001:** Clinical characteristics, comorbidities, and risk factors of the study population at baseline.

Parameters	
Age at time of scan, year, mean ± SD	60.43 ± 9.21
Male gender, *n* (%)	34 (65.38)
Body mass index (kg/m^2^), mean ± SD	28.57 ± 4.36
Time until the second CTA scan (days), mean ± SD	372.4 ± 68.06
Time until the last CTA scan (days), mean ± SD	1103 ± 108.4
LVEF ^1^ (%), mean ± SD	48.21 ± 5.37
Cardiovascular risk factors:	
Hypertension, *n* (%)	44 (84.61)
Hypercholesterolemia, *n* (%)	33 (63.46)
Diabetes mellitus, *n* (%)	14 (26.92)
Smoking, *n* (%)	9 (17.30)
Familial history of CAD ^2^, *n* (%)	22 (42.30)
Plaque types:	
Calcified plaque, *n* (%)	80 (54.06)
Non-calcified plaque, *n* (%)	7 (4.73)
Mixed plaque, *n* (%)	61 (41.21)
Lesion location:	
LAD ^3^, *n* (%)	87 (58.78)
LCX ^4^, *n* (%)	16 (10.81)
RCA ^5^, *n* (%)	45 (30.41)
Calcium score, mean ± SD	127.5 ± 72.96
<10, *n* (%)	3 (5.77)
10–400, *n* (%)	36 (69.23)
>400, *n* (%)	13 (25.00)

^1^ LVEF—left ventricular ejection fraction; ^2^ CAD—coronary artery disease; ^3^ LAD—left anterior descending artery; ^4^ LCX—left circumflex artery; ^5^ RCA—right coronary artery.

**Table 2 ijms-25-01700-t002:** Comparison of lipid panel outcomes across various follow-up appointments.

Lipid Profile Parameters	1st Visit	2nd Visit	*p*-Value *	Last Visit	*p*-Value
T-Cho ^1^ (mg/dL), mean ± SD, [95% CI]	194.3 ± 66.76 [176.1–212.5]	145.2 ± 34.7 [129.1–164.4]	0.0003	150.7 ± 48.92 [137.3–164.0]	<0.0001
LDL-Cho ^2^ (mg/dL), mean ± SD, [95% CI]	105.9 ± 33.97 [96.41–115.3]	87.69 ± 32.99 [78.50–96.87]	<0.0001	84.07 ± 32.16 [75.12–93.03]	<0.0001
HDL-Cho ^3^ (mg/dL), mean ± SD, [95% CI]	38.11 ± 8.72 [35.68–40.53]	47.10 ± 7.97 [44.88–49.32]	<0.0001	50.01 ± 6.46 [48.21–51.81]	<0.0001
TGs ^4^ (mg/dL), mean ± SD, [95% CI]	188.7 ± 66.01 [170.3–207.1]	179.4 ± 59.49 [162.9–196.0]	<0.0001	171.9 ± 53.76 [156.9–186.9]	<0.0001

^1^ T-Cho—total cholesterol; ^2^ LDL-Cho—low-density lipoprotein cholesterol; ^3^ HDL-Cho—high-density lipoprotein cholesterol; ^4^ TGs—triglycerides. * see the description in the statistical analysis section of Section 4.

**Table 3 ijms-25-01700-t003:** Comparison of plaque changes at the baseline and final follow-up appointments.

Serial Changes of Plaque Components	1st Scan	Last Scan	*p*-Value
Calcified plaques (*n* = 80)			
^1^ TPV (mm^3^), mean ± SD, [95% CI]	185.1 ± 78.70 [167.5–202.5]	206.8 ± 86.03 [187.7–226.0]	0.0773
Non-calcified plaques (*n* = 7)			
TPV (mm^3^), mean ± SD, [95% CI]	57.23 ± 26.96 [32.30–82.16]	55.51 ± 24.25 [33.09–77.94]	0.9122
^2^ FPV (mm^3^), mean ± SD, [95% CI]	47.36 ± 31.04 [18.65–76.06]	49.70 ± 25.14 [26.45–72.95]	0.8859
^3^ LRPV (mm^3^), mean ± SD, [95% CI]	5.14 ± 2.65 [2.68–7.60]	4.70 ± 3.34 [1.60–7.79]	0.2070
^4^ CPV (mm^3^), mean ± SD, [95% CI]	0	2.47 ± 1.89 [0.71–4.22]	0.0137
Mixed plaques (*n* = 61)			
TPV (mm^3^), mean ± SD, [95% CI]	237.4 ± 70.0 [219.5–255.4]	256.9 ± 79.84 [236.5–277.4]	0.1454
^5^ NCPV (mm^3^), mean ± SD, [95% CI]	180.5 ± 66.81 [163.3–197.6]	155.4 ± 59.51 [140.1–170.6]	0.0209

^1^ TPV—total plaque volume; ^2^ FPV—fibrotic plaque volume; ^3^ LRPV—lipid-rich plaque volume; ^4^ CPV—calcified plaque volume; ^5^ NCPV—non-calcified plaque volume.

## Data Availability

The data presented in this study are available on request from the corresponding author. The data are not publicly available due to privacy reasons.

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
