# Peer review of "Assessing the Impact of Long-Term High-Dose Statin Treatment on Pericoronary Inflammation and Plaque Distribution—A Comprehensive Coronary CTA Follow-Up Study"

_ijms, 2024, doi:10.3390/ijms25031700_

Round 1

Reviewer 1 Report

Comments and Suggestions for Authors

Very interesting validation of the pleiotropic effects of statins on the vasculature. A few minor comments:

1. Figure 4 and Table 4 present the same data. Please choose one. 

2. Figure 4 legend mentions lesion-specific analysis. Does this mean that the data presented are for plaques only? If so, which plaque was chosen, with what criteria? What happened if a vessel was plaque free?

3. If figure 6, which FAI SCORE and why was used? Wouldn't it be more relevant to use the Cari-Heart risk?

4. What are the clinical risk factors in Cari-Heart risk? These need to be mentioned as all analyses are unadjusted. 

5. I would suggest expanding the discussion with more contemporary data on imaging inflammation and plaques with radiomics. For reference: https://doi.org/10.1148/radiol.221693

and

 https://doi.org/10.1016/S2589-7500(22)00132-7

Author Response

We are truly grateful for the time you devoted to sharing your valuable insights on our recent study. It's encouraging to hear your interesting validation of the pleiotropic effects of statins on the vasculature. Your positive comments are greatly appreciated, and we are glad to know that our findings were seen as both significant and relevant.

Reviewer 2 Report

Comments and Suggestions for Authors

The paper IJMS 2832957 – » Assessing the impact of long-term high-dose statin treatment on pericoronary inflammation and plaque distribution - a comprehensive coronary CTA follow-up study« is a prospective observational study of 52 patients with CTA-documented coronary artery plaques causing 25-49% stenosis of at least one major coronary artery, who were prescribed statins at the beginning of the study. Follow-up coronary CTA was performed after 1 year and after 3 years. In mixed plaques, calcified portions increased in size and non-calcified portions decreased significantly. Pericoronary adipose tissue attenuation decreased after one year in LAD, decreased borderline significantly in LCX and insignificantly in RCA. The authors concluded that statins reduced pericoronary inflammation and plaque vulnerability.

The results are original and in line with previous studies.

Overall, this study focuses on the morphological characteristics of tissue and lacks detail in describing the clinical characteristics of patients.

 I have the following specific concerns:

 Abstract:

- Please, define all abbreviations.

- Line 22-23.  Sate that follow-up CTA was performed at approximately 1 year and 3 years after inclusion.

 Abstract, line 19, and Materials and Methods, line 274. Was chest pain always typical?

 Materials and Methods, or better »Patients and Methods«:

This section should be moved to section 2, preceding the Results.

- Please, describe how many patients were screened to find 117 patients with 25- 49 % coronary stenosis?

-Describe which statins were used and in which doses.

- Methods, lines 320-326. The CaRi Herat risk score should be described in more detail and referenced.

 Discussion. Why was the response of pericoronary adipose tissue most pronounced in LAD and less so in LCX and RCA? This should be briefly discussed.

 Figures. In the legends, describe all abbreviations used.

Author Response

We would like to express our utmost appreciation for the time you took to provide us with your valuable feedback on our recent study. Your acknowledgement of the originality and alignment of our results with previous studies is highly encouraging. We appreciate your observation regarding the study's emphasis on morphological tissue characteristics and acknowledge the suggestion to enhance the description of clinical patient characteristics. This feedback is invaluable, and we will consider incorporating more detailed clinical data to provide a holistic view of the study's impact. 

Round 2

Reviewer 2 Report

Comments and Suggestions for Authors

The authors have successfully addressed my concerns.